# RECURSIVE NEIGHBORHOOD POOLING FOR GRAPH REPRESENTATION LEARNING

## ABSTRACT

While massage passing based Graph Neural Networks (GNNs) have become increasingly popular architectures for learning with graphs, recent works have revealed important shortcomings in their expressive power. In response, several higher-order GNNs have been proposed, which substantially increase the expressive power, but at a large computational cost. Motivated by this gap, we introduce and analyze a new recursive pooling technique of local neighborhoods that allows different tradeoffs of computational cost and expressive power. First, we show that this model can count subgraphs of size $k$, and thereby overcomes a known limitation of low-order GNNs. Second, we prove that, in several cases, RNP-GNNs can greatly reduce computational complexity compared to the existing higher-order $k$-GNN and Local Relational Pooling (LRP) networks.

## 1 INTRODUCTION

Graph Neural Networks (GNNs) are powerful tools for graph representation learning (Scarselli et al., 2008; Kipf & Welling, 2017; Hamilton et al., 2017), and have been successfully used in applications such as encoding molecules, simulating physics, social network analysis, knowledge graphs, and many others (Duvenaud et al., 2015; Defferrard et al., 2016; Battaglia et al., 2016; Jin et al., 2018). An important class of GNNs is the set of Message Passing Graph Neural Networks (MPNNs) (Gilmer et al., 2017; Kipf & Welling, 2017; Hamilton et al., 2017; Xu et al., 2019; Scarselli et al., 2008), which follow an iterative message passing scheme to compute a graph representation.

Despite the empirical success of MPNNs, their expressive power has been shown to be limited. For example, their discriminative power, at best, corresponds to the one-dimensional Weisfeiler-Leman (1-WL) graph isomorphism test (Xu et al., 2019; Morris et al., 2019), so they cannot, e.g., distinguish regular graphs. Moreover, they also cannot count any induced subgraph with at least three vertices (Chen et al., 2020), or learn graph parameters such as clique information, diameter, or shortest cycle (Garg et al., 2020). Still, in several applications, e.g. in computational chemistry, materials design or pharmacy (Elton et al., 2019; Sun et al., 2020; Jin et al., 2018), we aim to learn functions that depend on the presence or count of specific substructures.

To strengthen the expressive power of GNNs, higher-order representations such as $k-$GNNs (Morris et al., 2019) and $k-$Invariant Graph Networks ($k-$IGNs) (Maron et al., 2019) have been proposed. $k-$GNNs are inspired by the $k$-dimensional WL ($k-$WL) graph isomorphism test, a message passing algorithm on $k-$tuples of vertices, and $k-$IGNs are based on equivariant linear layers of a feed-forward neural network applied to the input graph as a matrix, and are at least as powerful as $k-$GNN. These models are provably more powerful than MPNNs and can, e.g., count any induced substructure with at most $k$ vertices. But, this power comes at the computational cost of at least $\Omega(n^k)$ operations for $n$ vertices. The necessary tradeoffs between expressive power and computational complexity are still an open question.

The expressive power of a GNN is often measured in terms of a hierarchy of graph isomorphism tests, i.e., by comparing it to a $k-$WL test. Yet, there is limited knowledge about how the expressive power of higher-order graph isomorphism tests relates to various functions of interest (Arvind et al., 2020). A different approach is to take the perspective of specific functions that are of practical interest, and quantify a GNN's expressive power via those. Here, we focus on counting induced substructures to measure the power of a GNN, as proposed in (Chen et al., 2020). In particular, we

study whether it is possible to count given substructures with a GNN whose complexity is between that of MPNNs and the existing higher-order GNNs.

To this end, we study the scheme of many higher-order GNNs (Morris et al., 2019; Chen et al., 2020): select a collection of subgraphs of the input graph, encode these, and (possibly iteratively) compute a learned function on this collection. First, we propose a new such class of GNNs, Recursive Neighborhood Pooling Graph Neural Networks (RNP-GNNs). Specifically, RNP-GNNs represent each vertex by a representation of its neighborhood of a specific radius. Importantly, this neighborhood representation is computed *recursively* from its subgraphs. As we show, RNP-GNNs can count any induced substructure with at most $k$ vertices. Moreover, for any set of substructures with at most $k$ vertices, there is a specifiable RNP-GNN that can count them. This flexibility allows to design a GNN that is adapted to the power needed for the task of interest, in terms of counting (induced) substructures.

The Local Relational Pooling (LRP) architecture too has been introduced with the goal of counting substructures (Chen et al., 2020). While it can do so, it is polynomial-time only if the encoded neighborhoods are of size $o(\log(n))$. In contrast, RNP-GNNs use almost linear operations, i.e., $n^{1+o(1)}$, if the size of each encoded neighborhood is $n^{o(1)}$. This is an exponential theoretical improvement in the tolerable size of neighborhoods, and a significant improvement over the complexity of $O(n^k)$ in $k-$GNN and $k-$IGN.

Finally, we take a broader perspective and provide an information-theoretic lower bound on the complexity of a general class of GNNs that can provably count substructures with at most $k$ vertices. This class includes GNNs that represent a given graph by aggregating a number of encoded graphs, where the encoded graphs are related to the given graph with an arbitrary function.

In short, in this paper, we make the following contributions:

- We introduce Recursive Neighborhood Pooling Graph Neural Networks (RNP-GNNs), a flexible class of higher-order graph neural networks, that provably allow to design graph representation networks with any expressive power of interest, in terms of counting (induced) substructures.
- We show that RNP-GNNs offer computational gains over existing models that count substructures: an exponential improvement in terms of the "tolerable" size of the encoded neighborhoods compared to LRP networks, and much less complexity in sparse graphs compared to $k-$GNN and $k-$IGN.
- We provide an information-theoretic lower bound on the complexity of a general class of GNN that can count (induced) substructures with at most $k$ vertices.

## 2 BACKGROUND

**Message Passing Graph Neural Networks.** Let $G = (\mathcal{V}, \mathcal{E}, X)$ be a labeled graph with $|\mathcal{V}| = n$ vertices. Here, $X_v \in \mathcal{X}$ denotes the initial label of $v \in \mathcal{V}$, where $\mathcal{X} \subseteq \mathbb{N}$ is a (countable) domain.

A typical Message Passing Graph Neural Network (MPNN) first computes a representation of each vertex, and then aggregates the vertex representations via a readout function into a representation of the entire graph $G$. The representation $h_v^{(i)}$ of each vertex $v \in \mathcal{V}$ is computed iteratively by aggregating the representations $h_u^{(i-1)}$ of the neighboring vertices $u$:

$$m_v^{(i)} = \text{AGGREGATE}^{(i)}\left(\{\!\!\{h_u^{(i-1)} : u \in \mathcal{N}(v)\}\!\!\}\right), \tag{1}$$

$$h_v^{(i)} = \text{COMBINE}^{(i)}\left(h_v^{(i-1)}, m_v^{(i)}\right), \tag{2}$$

for any $v \in \mathcal{V}$, for $k$ iterations, and with $h_v^{(0)} = X_v$. The AGGREGATE/COMBINE functions are parametrized, learnable functions, and $\{\!\!\{.\}\!\!\}$ denotes a multi-set, i.e., a set with (possibly) repeating elements. A graph-level representation can be computed as

$$h_G = \text{READOUT}\left(h_v^{(k)} : v \in \mathcal{V}\right), \tag{3}$$

where READOUT is a learnable function. For representational power, it is important that the learnable functions above are injective, which can be achieved, e.g., if the AGGREGATE function is a summation and COMBINE is a weighted sum concatenated with an MLP (Xu et al. (2019)).

**Higher-Order GNNs.** To increase the representational power of GNNs, several higher-order GNNs have been proposed. In a $k-$*GNN*, a message passing algorithm is applied to the $k-$tuples of vertices, in a similar fashion as GNNs do on vertices (Morris et al., 2019). At initialization, each $k-$tuple is labeled with its type, that is, two $k-$tuples are labeled differently if their induced subgraphs are not isomorphic. As a result, $k-$GNNs can count (induced) substructures with at most $k$ vertices even at initialization. Another class of higher-order networks are $k-$*IGNs*, which are constructed with linear invariant/equivariant feed-forward layers, whose inputs consider graphs via adjacency matrices (Maron et al., 2019). $k-$IGNs are at least as powerful as $k-$GNNs, and hence they too can count substructures with at most $k$ vertices. However, both methods need $O(n^k)$ operations.

Specifically for counting substructures, Chen et al. (2020) propose *Local Relational Pooling (LRP)* networks. LRPs apply Relational Pooling (RP) networks (Murphy et al., 2019b;a) on the neighborhoods around each vertex. RP networks use permutation-variant functions and convert them to a permutation-invariant function by summing over all permutations. This summation is computationally expensive.

## 3 OTHER RELATED WORKS

**Expressive power.** Several other works have studied the expressive power of GNNs (Azizian & Lelarge, 2020). Scarselli et al. (2009) extend universal approximation from feedforward networks to GNNs, using the notion of *unfolding equivalence*. Chen et al. (2019) establish an equivalence between the graph isomorphism problem and the power to approximate permutation invariant functions on graphs. Maron et al. (2019) and Keriven & Peyré (2019) propose higher-order, tensor-based GNN models that provably achieve universal approximation of permutation-invariant functions on graphs, and Loukas (2019) studies expressive power under depth and width restrictions. Studying GNNs from the perspective of local algorithms, Sato et al. (2019) show that GNNs can approximate solutions to certain combinatorial optimization problems.

**Subgraphs and GNNs.** The idea of considering local neighborhoods to have better representations than MPNNs is considered in several works (Liu et al., 2019; Monti et al., 2018; Liu et al., 2020; Yu et al., 2020; Meng et al., 2018; Cotta et al., 2020; Alsentzer et al., 2020; Huang & Zitnik, 2020). For example, in link prediction, one can use local neighborhoods around links and apply GNNs, as suggested in (Zhang & Chen, 2018). A novel method based on combining GNNs and a clustering algorithm is proposed in (Ying et al. (2018)). For graph comparison (i.e., testing whether a given possibly large subgraph exists in the given model), Ying et al. (2020) compare the outputs of GNNs for small subgraphs of the two graphs. To improve the expressive power of GNNs, Bouritsas et al. (2020) use features that are counts of specific subgraphs of interest. Another related work is (Vignac et al., 2020), where an MPNN is strengthened by learning local context matrices around vertices.

## 4 RECURSIVE NEIGHBORHOOD POOLING

Next, we construct Recursive Neighborhood Pooling Graph Neural Networks (RNP-GNNs), GNNs that can count any set of induced substructures of interest, with lower complexity than previous models. We represent each vertex by a representation of its radius $r_1$-neighborhood, and then combine these representations. The key question is how to encode these local neighborhoods in a vector representation. To do so, we introduce a new idea: we view local neighborhoods as small subgraphs, and recursively apply our model to encode these neighborhood subgraphs. When encoding the local subgraphs, we use a different radius $r_2$, and, recursively, a sequence of radii $(r_1, r_2, \ldots, r_t) \in \mathbb{N}^t$ to obtain the final representation $h_v^{(t)}$ of vertices after $t$ recursion steps.

While MPNNs also encode a representation of a local neighborhood of certain radius, the recursive representations differ as they essentially take into account intersections of neighborhoods. As a result, as we will see in Section 5.1, they retain more structural information and are more expressive. Models such as $k$-GNN and LRP also compute encodings of subgraphs, and then update the resulting representations via message passing. We can do the same with the neighborhood representations computed by RNP-GNNs to encode more global information, although our representation results in Section 5.1 hold even without that. In Section 6, we will compare the computational complexity of RNP-GNN and these other models.

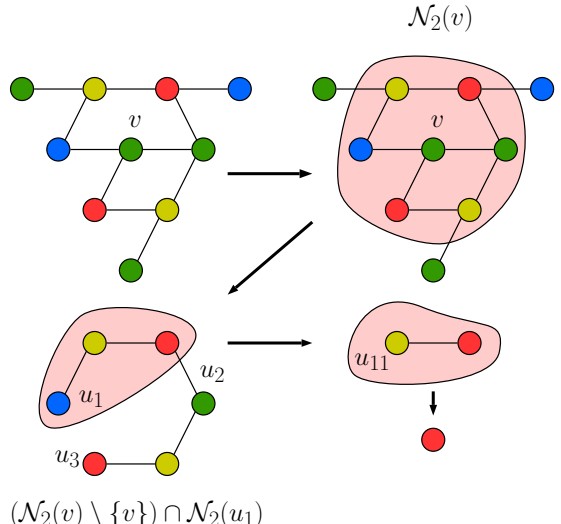

Figure 1: Illustration of a Recursive Neighborhood Pooling GNN (RNP-GNN) with recursion parameters $(2, 2, 1)$. To compute the representation of vertex $v$ in the given input graph (depicted in the top left of the figure), we first recurse on $G(\mathcal{N}_2(v) \setminus \{v\})$, depicted in the top right of the figure). To do so, we find the representation of each vertex $u \in G(\mathcal{N}_2(v) \setminus \{v\})$. For instance, to compute the representation of $u_1$, we apply an RNP-GNN with recursion parameters $(2, 1)$ and aggregate $G((\mathcal{N}_2(v) \setminus \{v\}) \cap (\mathcal{N}_2(u_1) \setminus \{u_1\}))$, which is shown in the bottom left of the figure. To do so, we recursively apply an RNP-GNN with recursion parameter $(1)$ on $G((\mathcal{N}_2(v) \setminus \{v\}) \cap (\mathcal{N}_2(u_1) \setminus \{u_1\}) \cap (\mathcal{N}_1(u_{11}) \setminus \{u_{11}\}))$, in the bottom right of the figure.

Formally, an RNP-GNN is a parametrized learnable function $f(.; \theta) : \mathbb{G}_n \to \mathbb{R}^d$, where $\mathbb{G}_n$ is the set of all labeled graphs on $n$ vertices. Let $G = (\mathcal{V}, \mathcal{E}, X)$ be a labeled graph with $|\mathcal{V}| = n$ vertices, and let $h_v^{(0)} = X_v$ be the initial representation of each vertex $v$. Let $\mathcal{N}_r(v)$ denote the neighborhood of radius $r$ of vertex $v$, and let $G_v^{(t-1)}\left(\mathcal{N}_{r_1}(v) \setminus \{v\}\right)$ denote the induced subgraph of $G$ on the set of vertices $\mathcal{N}_{r_1}(v) \setminus \{v\}$, with augmented vertex label $X_u^{(t-1)} = (h_u^{(t-1)}, \mathbb{1}[(u, v) \in \mathcal{E}])$ for any $u \in \mathcal{N}_{r_1}(v) \setminus \{v\}$. This means we add information about whether vertices are direct neighbors (with distance one) of $v$. Given a recursion sequence $(r_1, r_2, \ldots, r_t)$ or radii, the representations are updated as

$$m_v^{(t)} = \text{RNP-GNN}^{(t-1)}\left(G_v^{(t-1)}(\mathcal{N}_{r_1}(v) \setminus \{v\})\right), \qquad (4)$$

$$h_v^{(t)} = \text{COMBINE}^{(t)}\left(h_v^{(t-1)}, m_v^{(t)}\right), \qquad (5)$$

for any $v \in \mathcal{V}$, and

$$h_G = \text{READOUT}\left(\{\!\{h_v^{(t)} : v \in \mathcal{V}\}\!\}\right). \qquad (6)$$

Different from MPNNs, the recursive update (4) is in general applied to a subgraph, and not a multi-set of vertex representations. RNP-GNN$^{(t-1)}$ is an RNP-GNN with recursion parameters $(r_2, \ldots, r_t) \in \mathbb{N}^{t-1}$. The final READOUT is an injective, permutation-invariant learnable multi-set function.

If $t = 1$, then

$$m_v^{(t)} = \text{AGGREGATE}^{(t)}\left(\{\!\{(h_u^{(t-1)}, \mathbb{1}[(u, v) \in \mathcal{E}]) : u \in \mathcal{N}_{r_1}(v)\}\!\}\right) \qquad (7)$$

is a permutation-invariant aggregation function as used in MPNNs, only over a potentially larger neighborhood. For $r_1 = 1$ and $t = 1$, RNP-GNN reduces to MPNN.

In Figure 1, we illustrate an RNP-GNN with recursion parameters $(2, 2, 1)$ as an example. We also provide pseudocode for RNP-GNNs in Appendix C.

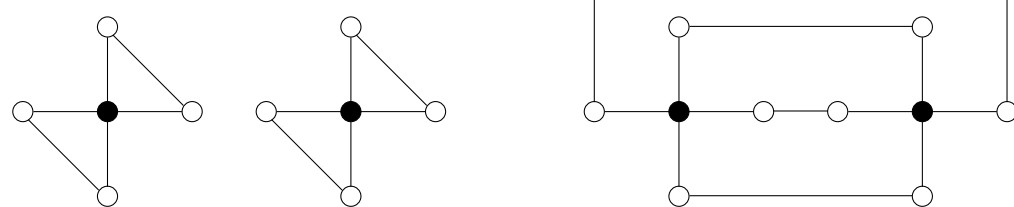

Figure 2: MPNNs cannot count substructures with three vertices or more (Chen et al., 2020). For example, the graph with black center vertex on the left cannot be counted, since the two graphs on the left result in the same vertex representations as the graph on the right.

## 5 EXPRESSIVE POWER

In this section, we analyze the expressive power of RNP-GNNs.

### 5.1 COUNTING (INDUCED) SUBSTRUCTURES

In contrast to MPNNs, which, in general, cannot count substructures of three vertices or more, in this section we prove that for any set of substructures, there is an RNP-GNN that provably counts them. We begin with a few definitions.

**Definition 1.** *Let $G, H$ be arbitrary labeled simple graphs, where $\mathcal{V}$ is the set of vertices in $G$. Also, for any $\mathcal{S} \subseteq \mathcal{V}$, let $G(\mathcal{S})$ denote the subgraph of $G$ induced by $\mathcal{S}$. The* induced subgraph count function *is defined as*

$$C(G; H) := \sum_{\mathcal{S} \subseteq \mathcal{V}} \mathbb{1}\{G(\mathcal{S}) \cong H\}, \tag{8}$$

*i.e., the number of subgraphs of $G$ isomorphic to $H$. For unlabeled $H$, the function is defined analogously.*

We also need to define a notion of *covering* for graphs. Our definition uses distances on graphs.

**Definition 2.** *Let $H = (\mathcal{V}_H, \mathcal{E}_H)$ be a (possibly labeled) simple connected graph. For any $\mathcal{S} \subseteq \mathcal{V}_H$ and $v \in \mathcal{V}_H$, define*

$$\bar{d}_H(v; \mathcal{S}) := \max_{u \in \mathcal{S}} d(u, v), \tag{9}$$

*where $d(.,.)$ is the shortest-path distance in $H$.*

**Definition 3.** *Let $H$ be a (possibly labeled) simple connected graph on $t + 1$ vertices. A permutation of vertices, such as $(v_1, v_2, \ldots, v_{t+1})$, is called a* vertex covering sequence, *with respect to a sequence $\mathbf{r} = (r_1, r_2, \ldots, r_t) \in \mathbb{N}^t$ called a* covering sequence, *if and only if*

$$\bar{d}_{H'_i}(v_i; \mathcal{S}_i) \leq r_i, \tag{10}$$

*for any $i \in [t + 1] = \{1, 2, \ldots, t + 1\}$, where $\mathcal{S}_i = \{v_i, v_{i+1} \ldots, v_{t+1}\}$ and $H'_i = H(\mathcal{S}_i)$ is the subgraph of $H$ induced by the set of vertices $\mathcal{S}_i$. We also say that $H$ admits* the covering sequence *$\mathbf{r} = (r_1, r_2, \ldots, r_t) \in \mathbb{N}^t$ if there is a vertex covering sequence for $H$ with respect to $\mathbf{r}$.*

In particular, in a covering sequence we first consider the whole graph as a local neighborhood of one of its vertices with radius $r_1$. Then, we remove that vertex and compute the covering sequence of the remaining graph. Figure 3 shows an example of a covering sequence computation. An important property, which holds by definition, is that if $\mathbf{r}$ is a covering sequence for $H$, then any $\mathbf{r}' \geq \mathbf{r}$ (in a point-wise sense) is also a covering sequence for $H$.

Note that any connected graph on $k$ vertices admits at least one covering sequence, which is $(k - 1, k - 2, \ldots, 1)$. To observe this fact, note that in a connected graph, there is at least one vertex that can be removed and the remaining graph still remains connected. Therefore, we may take this vertex as the first element of a vertex covering sequence, and inductively find the other elements. Since the diameter of a connected graph with $k$ vertices is always bounded by $k - 1$, we achieve the desired

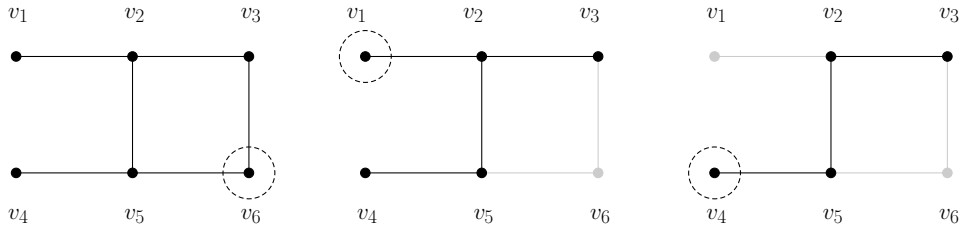

Figure 3: Example of a covering sequence computed for the graph on the left. For this graph, $(v_6, v_1, v_4, v_5, v_3, v_2)$ is a vertex covering sequence with respect to the covering sequence $(3, 3, 3, 2, 1)$. The first two computations to obtain this covering sequence are depicted in the middle and on the right.

result. However, we will see in the next section that, when using covering sequences to identify sufficiently powerful RNP-GNNs, it is desirable to have covering sequences with low $r_1$, since the complexity of the resulting RNP-GNN depends on $r_1$. We provide an algorithm in Appendix D to find such covering sequences in polynomial time

More generally, if $H_1$ and $H_2$ are (possibly labeled) simple graphs on $k$ vertices and $H_1 \Subset H_2$, i.e., $H_1$ is a subgraph of $H_2$ (not necessarily induced-subgraph), then, it follows from the definition that any covering sequence for $H_1$ is also a covering sequence for $H_2$. As a side remark, as illustrated in Figure 4, covering sequences need not always to be decreasing.

Using covering sequences, we can show the following result.

**Theorem 1.** *Consider a set of (labeled or unlabeled) graphs $\mathcal{H}$ on $t + 1$ vertices, such that any $H \in \mathcal{H}$ admits the covering sequence $(r_1, r_2, \ldots, r_t)$. Then, there is an RNP-GNN with recursion parameters $(r_1, r_2, \ldots, r_t)$ that can count any $H \in \mathcal{H}$. In other words, if there exists $H \in \mathcal{H}$ such that $C(G_1; H) \neq C(G_2; H)$, then $f(G_1; \theta) \neq f(G_2; \theta)$. The same result also holds for the non-induced subgraph count function.*

Theorem 1 states that, with appropriate recursion parameters, any set of (labeled or unlabeled) substructures can be counted by an RNP-GNN. Interestingly, induced and non-induced subgraphs can be both counted in RNP-GNNs[1].

The theorem holds for any covering sequence that is valid for all graphs in $\mathcal{H}$. For any graph, one can compute a covering sequence by computing a spanning tree, and sequentially pruning the leaves of the tree. The resulting sequence of nodes is a vertex covering sequence, and the corresponding covering sequence can be obtained from the tree too (Appendix D). A valid covering sequence for all the graphs in $\mathcal{H}$ is the coordinate-wise maximum of all these sequences.

For large substructures, the sequence $(r_1, r_2, \ldots, r_t)$ can be long or include large numbers, and this will affect the computational complexity of RNP-GNNs. For small, e.g., constant-size substructures, the recursion parameters are also small (i.e., $r_i = O(1)$ for all $i$), raising the hope to count these structures efficiently. In particular, $r_1$ is an important parameter. In Section 6, we analyze the complexity of RNP-GNNs in more detail.

## 5.2 A Universal Approximation Result for Local Functions

Theorem 1 shows that RNP-GNNs can count substructures if their recursion parameters are chosen carefully. Next, we provide a universal approximation result, which shows that they can learn any function related to local neighborhoods or small subgraphs in a graph.

First, we recall that for a graph $G$, $G(\mathcal{S})$ denotes the subgraph of $G$ induced by the set of vertices $\mathcal{S}$.

**Definition 4.** *A function $\ell : \mathbb{G}_n \to \mathbb{R}^d$ is called an $r-$local graph function if*

$$\ell(G) = \phi(\{\!\{\psi(G(\mathcal{S})) : \mathcal{S} \subseteq \mathcal{V}, |\mathcal{S}| \leq r\}\!\}), \tag{11}$$

*where $\psi : \mathbb{G}_r \to \mathbb{R}^{d'}$ is a function on graphs and $\phi$ is a multi-set function.*

---

[1]For simplicity, we assume that $\mathcal{H}$ only contains $t + 1$ vertex graphs. If $\mathcal{H}$ includes graphs with strictly less than $t + 1$ vertices, we can simply add a sufficient number of zeros to the RHS of their covering sequences.

In other words, a local function only depends on small substructures.

**Theorem 2.** *For any $r-$local graph function $\ell(.)$, there exists an RNP-GNN $f(.;\theta)$ with recursion parameters $(r-1, r-2, \ldots, 1)$ such that $f(G; \theta) = \ell(G)$ for any $G \in \mathbb{G}_n$.*

As a result, we can provably learn all the local information in a graph with an appropriate RNP-GNN. Note that we still need recursions, because the function $\psi(.)$ may be an arbitrarily difficult graph function. However, to achieve the full generality of such a universal approximation result, we need to consider large recursion parameters ($r_1 = r-1$) and injective aggregations in the RNP-GNN network. For universal approximation, we may also need high dimensions if feedforward network layers are used for aggregation (see the proof of the theorem for more details).

As a remark, for $r = n$, achieving universal approximation on graphs implies solving the graph isomorphism problem. But, in this extreme case, the computational complexity of the model in general is not a polynomial in $n$.

## 6 COMPUTATIONAL COMPLEXITY

The computational complexity of RNP-GNNs is graph-dependent. For instance, we need to compute the set of local neighborhoods, which is cheaper for sparse graphs. A complexity measure existing in the literature is the tensor order. For higher-order networks, e.g., $k-$IGNs, we need to consider tensors in $\mathbb{R}^{n^k}$. The space complexity is then $O(n^k)$ and the time complexity can be even more, dependent on the algorithm used to process tensors. In general, for a message passing algorithm on graphs, the complexity of the model depends linearly on the number of vertices (if the graph is sparse). Therefore, to bound the complexity of a method, we need to bound the number of node representation updates, which we do in the following theorem.

**Theorem 3.** *Let $f(.;\theta) : \mathbb{G}_n \to \mathbb{R}^d$ be an RNP-GNN with the recursion parameters $(r_1, r_2, \ldots, r_t)$. Assume that the observed graphs $G_1, G_2, \ldots$, whose representations we compute, satisfy the following property:*

$$\max_{v \in [n]} |\mathcal{N}_{r_1}(v)| \leq c, \tag{12}$$

*where $c$ is a graph independent constant. Then, the number of node updates in the RNP-GNN is $O(nc^t)$.*

In other words, if $c = n^{o(1)}$ and $t = O(1)$, then RNP-GNN requires relatively few updates (that is, $n^{1+o(1)}$), compared to the higher-order networks ($O(n^{t+1})$). Also, in this case, finding neighborhoods is not difficult, since neighborhoods are small ($n^{o(1)}$). Note that if the maximum degree of the given graphs is $\Delta$, then $c = O(r_1\Delta^{r_1})$. Therefore, similarly, if $\Delta = n^{o(1)}$ then we can count with at most $n^{1+o(1)}$ updates.

The above results show that when using RNP-GNNs with sparse graphs, we can learn functions of substructures with $k$ vertices without requiring $k-$order tensors. LRPs also encode neighborhoods of distance $r_1$ around nodes. In particular, all $c!$ permutations of the nodes in a neighborhood of size $c$ are considered to obtain the representation. As a result, LRP networks only have polynomial complexity if $c = o(\log(n))$. Thus, RNP-GNNs can provide an exponential improvement in terms of the tolerable size $c$ of neighborhoods with distance $r_1$ in the graph.

Moreover, theorem 3 suggests to aim for small $r_1$. The other $r_i$'s may be larger than $r_1$, as shown in Figure 4, but do not affect the upper bound on the complexity.

## 7 AN INFORMATION-THEORETIC LOWER BOUND

In this section, we provide a general information-theoretic lower bound for graph representations that encode a given graph $G$ by first encoding a number of (possibly small) graphs $G_1, G_2, \ldots, G_t$ and then aggregating the resulting representations. The sequence of graphs $G_1, G_2, \ldots, G_t$ may be obtained in an arbitrary way from $G$. For example, in an MPNN, $G_i$ can be the computation tree (rooted tree) at node $i$. As another example, in LRP, $G_i$ is the local neighborhood around node $i$.

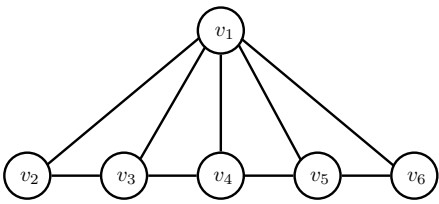

Figure 4: For the above graph, $(v_1, v_2, \ldots, v_6)$ is a vertex covering sequence. The corresponding covering sequence $(1, 4, 3, 2, 1)$ is not decreasing.

Formally, consider a graph representation $f(.; \theta) : \mathbb{G}_n \to \mathbb{R}^d$ as

$$f(G; \theta) = \Phi(\{\!\{\psi(G_i) : i \in [t]\}\!\}), \quad [t] = \{1, \ldots, t\} \tag{13}$$

for any $G \in \mathbb{G}_n$, where $\Phi$ is a multi-set function, $(G_1, G_2, \ldots, G_t) = \Xi(G)$ where $\Xi(.) : \mathbb{G}_n \to \left(\bigcup_{m=1}^{\infty} \mathbb{G}_m\right)^t$ is a function from one graph to $t$ graphs, and $\psi : \bigcup_{m=1}^{\infty} \mathbb{G}_m \to [s]$ is a function on graphs taking $s$ values. In short, we encode $t$ graphs, and each encoding takes one of $s$ values. We call this graph representation function an $(s, t)$-good graph representation.

**Theorem 4.** *Consider a parametrized class of $(s, t)-$good representations $f(.; \theta) : \mathbb{G}_n \to \mathbb{R}^d$ that is able to count any (not necessarily induced[2]) substructure with $k$ vertices. More precisely, for any graph $H$ with $k$ vertices, there exists $f(.; \theta)$ such that if $C(G_1; H) \neq C(G_2; H)$, then $f(G_1; \theta) \neq f(G_2; \theta)$. Then[3], $t = \tilde{\Omega}(n^{\frac{k}{s-1}})$.*

In particular, for any $(s, t)-$good graph representation with $s = 2$, i.e., binary encoding functions, we need $\tilde{\Omega}(n^k)$ encoded graphs. This implies that, for $s = 2$, enumerating all subgraphs and deciding for each whether it equals $H$ is near optimal. Moreover, if $s \leq k$, then $t = \Omega(n)$ small graphs would not suffice to enable counting.

More interestingly, if $k, s = O(1)$, then it is impossible to perform the substructure counting task with $t = O(\log(n))$. As a result, in this case, considering $n$ encoded graphs (as is done in GNNs or LRP networks) cannot be exponentially improved.

The lower bound in this section is information-theoretic and hence applies to any algorithm. It may be possible to strengthen it by considering computational complexity, too. For binary encodings, i.e., $s = 2$, however, we know that the bound cannot be improved since manual counting of subgraphs matches the lower bound.

## 8 TIME COMPLEXITY LOWER BOUNDS FOR COUNTING SUBGRAPHS

In this section, we put our results in the context of known hardness results for subgraph counting.

In general, the subgraph isomorphism problem is known to be NP-complete. Going further, the Exponential Time Hypothesis (ETH) is a conjecture in complexity theory (Impagliazzo & Paturi, 2001), and states that several NP-complete problems cannot be solved in sub-exponential time. ETH, as a stronger version of the $P \neq NP$ problem, is widely believed to hold. Assuming that ETH holds, the $k-$clique detection problem requires at least $n^{\Omega(k)}$ time (Chen et al., 2005). This means that if a graph representation can count *any* subgraph $H$ of size $k$, then computing it requires at least $n^{\Omega(k)}$ time.

**Corollary 1.** *Assuming ETH conjecture holds, any graph representation that can count any substructure $H$ on $k$ vertices with appropriate parametrization needs $n^{\Omega(k)}$ time to compute.*

The above bound matches the $O(n^k)$ complexity of the higher-order GNNs. Comparing with Theorem 4 above, Corollary 1 is more general, while Theorem 4 has fewer assumptions and offers a refined result for aggregation-based graph representations.

---

[2]The theorem also holds for induced-subgraphs, with/without vertex labels.

[3]$\tilde{\Omega}(m)$ is $\Omega(m)$ up to poly-logarithmic factors.

Given that Corollary 1 is a *worst-case* bound, a natural question is whether we can do better for subclasses of graphs. Regarding $H$, even if $H$ is a random Erdös-Rényi graph, it can only be counted in $n^{\Omega(k/\log k)}$ time (Dalirrooyfard et al., 2019).

Regarding the input graph in which we count, consider two classes of sparse graphs: *strongly sparse graphs* have maximum degree $\Delta = O(1)$, and *weakly sparse graphs* have average degree $\bar{\Delta} = O(1)$. We argued in Theorem 3 that RNP-GNNs achieve almost *linear* complexity for the class of strongly sparse graphs. For weakly sparse graphs, in contrast, the complexity of RNP-GNNs is generally not linear, but still polynomial, and can be much better than $O(n^k)$. One may ask whether it is possible to achieve a learnable graph representation such that its complexity for weakly sparse graphs is still linear. Recent results in complexity theory imply that this is impossible:

**Corollary 2** (Gishboliner et al. (2020); Bera et al. (2019)). *There is no graph representation algorithm that runs in linear time on weakly sparse graphs and is able to count any substructure $H$ on $k-$vertices (with appropriate parametrization).*

Hence, RNP-GNNs are close to optimal for several cases of counting substructures with parametrized learnable functions.

## 9 CONCLUSION

In this paper, we studied the theoretical possibility of counting substructures (induced-subgraphs) by a graph representation network. We proposed an architecture, called RNP-GNN, and we proved that for reasonably sparse graphs we can efficiently count substructures. Characterizing the expressive power of GNNs via the set of functions they can learn on substructures may be useful for developing new architectures. In the end, we proved a general lower bound for any graph representation which counts subgraphs and works by aggregating representations of a collection of graphs derived from the graph.

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

# A PROOFS

## A.1 PROOF OF THEOREM 1

### A.1.1 PRELIMINARIES

Let us first state a few definitions about the graph functions. Note that for any graph function $f : \mathbb{G}_n \to \mathbb{R}^d$, we have $f(G) = f(H)$ for any $G \cong H$.

**Definition 5.** *Given two graph functions $f, g : \mathbb{G}_n \to \mathbb{R}^d$, we write $f \sqsupseteq g$, if and only if for any $G_1, G_2 \in \mathbb{G}_n$,*

$$\forall G_1, G_2 \in G_n : g(G_1) \neq g(G_2) \implies f(G_1) \neq f(G_2), \tag{14}$$

*or, equivalently,*

$$\forall G_1, G_2 \in G_n : f(G_1) = f(G_2) \implies g(G_1) = g(G_2). \tag{15}$$

**Proposition 1.** *Consider graph functions $f, g, h : \mathbb{G}_n \to \mathbb{R}^d$ such that $f \sqsupseteq g$ and $g \sqsupseteq h$. Then, $f \sqsupseteq h$. In other words, $\sqsupseteq$ is transitive.*

*Proof.* The proposition holds by definition. □

**Proposition 2.** *Consider graph functions $f, g : \mathbb{G}_n \to \mathbb{R}^d$ such that $f \sqsupseteq g$. Then, there is a function $\xi : \mathbb{R}^d \to \mathbb{R}^d$ such that $\xi \circ f = g$.*

*Proof.* Let $\mathbb{G}_n = \sqcup_{i \in \mathbb{N}} \mathcal{F}_i$ be the partitioning induced by the equality relation with respect to the function $f$ on $\mathbb{G}_n$. Similarly define $\mathcal{G}_i, i \in \mathbb{N}$ for $g$. Note that due to the definition, $\{\mathcal{F}_i : i \in \mathbb{N}\}$ is a refinement for $\{\mathcal{G}_i : i \in \mathbb{N}\}$. Define $\xi$ to be the unique mapping from $\{\mathcal{F}_i : i \in \mathbb{N}\}$ to $\{\mathcal{G}_i : i \in \mathbb{N}\}$ which respects the equality relation. One can observe that such $\xi$ satisfies the requirement in the proposition. □

**Definition 6.** *An RNP-GNN is called maximally expressive, if and only if*

- *all the aggregate functions are injective as mappings from a multi-set on a countable ground set to their codomain.*

- *all the combine functions are injective mappings.*

**Proposition 3.** *Consider two RNP-GNNs $f, g$ with the same recursion parameters $\mathbf{r} = (r_1, r_2, \ldots, r_t)$ where $f$ is maximally expressive. Then, $f \sqsupseteq g$.*

*Proof.* The proposition holds by definition. □

**Proposition 4.** *Consider a sequence of graph functions $f, g_1, \ldots, g_k$. If $f \sqsupseteq g_i$ for all $i \in [k]$, then*

$$f \sqsupseteq \sum_{i=1}^{k} c_i g_i, \tag{16}$$

*for any $c_i \in \mathbb{R}$, $i \in \mathbb{N}$.*

*Proof.* Since $f \sqsupseteq g_i$, we have

$$\forall G_1, G_2 \in G_n : f(G_1) = f(G_2) \implies g_i(G_1) = g_i(G_2), \tag{17}$$

for all $i \in [k]$. This means that for any $G_1, G_2 \in \mathbb{G}_n$ if $f(G_1) = f(G_2)$ then $g_i(G_1) = g_i(G_2)$, $i \in [k]$, and consequently $\sum_{i=1}^{k} c_i g_i(G_1) = \sum_{i=1}^{k} c_i g_i(G_2)$. Therefore, from the definition we conclude $f \sqsupseteq \sum_{i=1}^{k} c_i g_i$. Note that the same proof also holds in the case of countable summations as long as the summation is bounded. □

**Definition 7.** *Let $H = (\mathcal{V}_H, \mathcal{E}_H, X^H)$ be a labeled connected simple graph on $k$ vertices. For any labeled graph $G = (\mathcal{V}_G, \mathcal{E}_G, X^G) \in \mathbb{G}_n$, the induced subgraph count function $C(G; H)$ is defined as*

$$C(G; H) := \sum_{\mathcal{S} \subseteq [n]} \mathbb{1}\{G(S) \cong H\}. \tag{18}$$

*Also, let $\bar{C}(G; H)$ denote the number of non-induced subgraphs of $G$ which are isomorphic to $H$. It can be defined with the homomorphisms from $H$ to $G$. Formally, if $n > k$ define*

$$\bar{C}(G; H) := \sum_{\substack{\mathcal{S} \subseteq [n] \\ |\mathcal{S}| = k}} \bar{C}(G(\mathcal{S}); H). \tag{19}$$

*Otherwise, $n = k$, and we define*

$$\bar{C}(G; H) := \sum_{\tilde{H} \in \tilde{\mathcal{H}}(H)} c_{\tilde{H}, H} \times \mathbb{1}\{G \cong \tilde{H}\}, \tag{20}$$

*where*

$$\tilde{\mathcal{H}}(H) := \{\tilde{H} \in \mathbb{G}_k : \tilde{H} \ni H\}, \tag{21}$$

*is defined with respect to the graph isomorphism, and $c_{\tilde{H}, H} \in \mathbb{N}$ denotes the number of subgraphs in $H$ identical to $\tilde{H}$. Note that $\tilde{\mathcal{H}}(H)$ is a finite set and $\ni$ denotes being a (not necessarily induced) subgraph.*

**Proposition 5.** *Let $\mathcal{H}$ be a family of graphs. If for any $H \in \mathcal{H}$, there is an RNP-GNN $f_H(.; \theta)$ with recursion parameters $(r_1, r_2, \ldots, r_t)$ such that $f_H \sqsupseteq C(G; H)$, then there exists an RNP-GNN $f(.; \theta)$ with recursion parameters $(r_1, r_2, \ldots, r_t)$ such that $f \sqsupseteq \sum_{H \in \mathcal{H}} C(G; H)$.*

*Proof.* Let $f(.; \theta)$ be a maximally expressive RNP-GNN. Note that by the definition $f \sqsupseteq f_H$ for any $H \in \mathcal{H}$. Since $\sqsupseteq$ is transitive, $f \sqsupseteq C(G; H)$ for all $H \in \mathcal{H}$, and using Proposition 4, we conclude that $f \sqsupseteq \sum_{H \in \mathcal{H}} C(G; H)$. $\qquad\square$

The following proposition shows that there is no difference between counting induced labeled graphs and counting induced unlabeled graphs in RNP-GNNs.

**Proposition 6.** *Let $H_0$ be an unlabeled connected graph. Assume that for any labeled graph $H$, which is constructed by adding arbitrary labels to $H_0$, there exists an RNP-GNN $f_H(.; \theta_H)$ such that $f_H \sqsupseteq C(G; H)$, then for its unlabeled counterpart $H_0$, there exists an RNP-GNN $f(.; \theta)$ with the same recursion parameters as $f_H(.; \theta_H)$ such that $f \sqsupseteq C(G; H_0)$.*

*Proof.* If there exists an RNP-GNN $f_H(.; \theta_H)$ such that $f_H \sqsupseteq C(G; H)$, then for a maximally expressive RNP-GNN $f(.; \theta)$ with the same recursion parameters as $f_H$ we also have $f_H \sqsupseteq C(G; H)$. Let $\mathcal{H}$ be the set of all labeled graphs $H = (\mathcal{V}, \mathcal{E}, X) \in \mathbb{G}_k$ up to graph isomorphism, where $X \in \mathcal{X}^k$ for a countable set $\mathcal{X}$. Note that $\mathcal{H} = \{H_1, H_2, \ldots\}$ is a countable set. Now we write

$$C(G; H_0) = \sum_{\substack{\mathcal{S} \subseteq [n] \\ |\mathcal{S}| = k}} \mathbb{1}\{G(S) \cong H_0\} \tag{22}$$

$$= \sum_{\substack{\mathcal{S} \subseteq [n] \\ |\mathcal{S}| = k}} \sum_{i \in \mathbb{N}} \mathbb{1}\{G(S) \cong H_i\} \tag{23}$$

$$= \sum_{i \in \mathbb{N}} \sum_{\substack{\mathcal{S} \subseteq [n] \\ |\mathcal{S}| = k}} \mathbb{1}\{G(S) \cong H_i\} \tag{24}$$

$$= \sum_{i \in \mathbb{N}} C(G; H_i). \tag{25}$$

$$\tag{26}$$

Now using Proposition 4 we conclude that $f \sqsupseteq C(G; H_0)$ since $C(G; H_0)$ is always finite. $\qquad\square$

**Definition 8.** *Let $H$ be a (possibly labeled) simple connected graph. For any $\mathcal{S} \subseteq \mathcal{V}_H$ and $v \in \mathcal{V}_H$, define*

$$\bar{d}_H(v; \mathcal{S}) := \max_{u \in \mathcal{S}} d(u, v). \tag{27}$$

**Definition 9.** *Let $H$ be a (possibly labeled) connected simple graph on $k = t + 1$ vertices. A permutation of vertices, such as $(v_1, v_2, \ldots, v_{t+1})$, is called a vertex covering sequence, with respect to a sequence $\mathbf{r} = (r_1, r_2, \ldots, r_t) \in \mathbb{N}^t$, called a covering sequence, if and only if*

$$\bar{d}_{H_i'}(v_i; \mathcal{S}_i) \leq r_i, \tag{28}$$

*for $i \in [t+1]$, where $H_i' = H(\mathcal{S}_i)$ and $\mathcal{S}_i = \{v_i, v_{i+1}, \ldots, v_{t+1}\}$. Let $\mathcal{C}_H(\mathbf{r})$ denote the set of all vertex covering sequences with respect to the covering sequence $\mathbf{r}$ for $H$.*

**Proposition 7.** *For any $G, H \in \mathbb{G}_k$, if $G \sqsupseteq H$ (non-induced subgraph), then*

$$\mathcal{C}_H(\mathbf{r}) \subseteq \mathcal{C}_G(\mathbf{r}), \tag{29}$$

*for any sequence $\mathbf{r}$.*

*Proof.* The proposition follows from the fact that the function $\bar{d}$ is decreasing with introducing new edges. ∎

**Proposition 8.** *Assume that Theorem 1 holds for induced-subgraph count functions. Then, it also holds for the non-induced subgraph count functions.*

*Proof.* Assume that for a connected (labeled or unlabeled) graph $H$, there exists an RNP-GNN with appropriate recursion parameters $f_H(.; \theta_H)$ such that $f_H \sqsupseteq C(G; H)$, then we prove there exists an RNP-GNN $f(.; \theta)$ with the same recursion parameters as $f_H$ such that $f \sqsupseteq \bar{C}(G; H)$.

If there exists an RNP-GNN $f_H(.; \theta_H)$ such that $f_H \sqsupseteq C(G; H)$, then for a maximally expressive RNP-GNN $f(.; \theta)$ with the same recursion parameters as $f_H$ we also have $f \sqsupseteq C(G; H)$. Note that

$$\bar{C}(G, H) = \sum_{\substack{\mathcal{S} \subseteq [n] \\ |\mathcal{S}| = k}} \bar{C}(G(\mathcal{S}); H) \tag{30}$$

$$= \sum_{\substack{\mathcal{S} \subseteq [n] \\ |\mathcal{S}| = k}} \sum_{\tilde{H} \in \tilde{\mathcal{H}}(H)} c_{\tilde{H}, H} \times \mathbb{1}\{G(S) \cong \tilde{H}\} \tag{31}$$

$$= \sum_{\tilde{H} \in \tilde{\mathcal{H}}(H)} c_{\tilde{H}, H} \sum_{\substack{\mathcal{S} \subseteq [n] \\ |\mathcal{S}| = k}} \mathbb{1}\{G(S) \cong \tilde{H}\} \tag{32}$$

$$= \sum_{i \in \mathbb{N}} c_{H_i, H} \times C(G, H_i), \tag{33}$$

where $\tilde{\mathcal{H}}(H) = \{H_1, H_2, \ldots\}$.

**Claim 1.** $f \sqsupseteq C(G, H_i)$ *for any $i$.*

Using Proposition 4 and Claim 1 we conclude that $f \sqsupseteq \bar{C}(G; H)$ since $\bar{C}(G; H)$ is finite and $f \sqsupseteq C(G, H_i)$ for any $i$, and the proof is complete. The missing part which we must show here is that for any $H_i$ the sequence $(r_1, r_2, \ldots, r_t)$ which covers $H$ also covers $H_i$. This follows from Proposition 7. We are done. ∎

At the end of this part, let us introduce an important notation. For any labeled connected simple graph on $k$ vertices $G = (\mathcal{V}, \mathcal{E}, X)$, let $G_v^*$ be the resulting induced graph obtained after removing $v \in \mathcal{V}$ from $G$ with the new labels defined as

$$X_u^* := (X_u, \mathbb{1}\{(u, v) \in \mathcal{E}\}), \tag{34}$$

for each $u \in \mathcal{V} \setminus \{v\}$. We may also use $X_u^{*v}$ for more clarification.

A.1.2 PROOF OF THEOREM 1

We utilize an inductive prove on $t$, which is the length of the covering sequence of $H$. Equivalently, due to the definition, $t = k - 1$, where $k$ is the number of vertices in $H$. First, we note that due to Proposition 8, without loss of generality, we can assume that $H$ is a simple connected labeled graph and the goal is to achieve the induced-subgraph count function via an RNP-GNN with appropriate recursion parameters. We also consider only maximally expressive networks here to prove the desired result.

**Induction base.** For the induction base, i.e., $t = 1$, $H$ is a two-vertex graph. This means that we only need to count the number of a specific (labeled) edge in the given graph $G$. Note that in this case we apply an RNP-GNN with recursion parameter $r_1 \geq 1$. Denote the two labels of the vertices in $H$ by $X_1^H, X_2^H \in \mathcal{X}$. The output of an RNP-GNN $f(.; \theta)$ is

$$f(G; \theta) = \phi(\{\!\!\{\psi(X_v^G, \varphi(\{\!\!\{X_u^{*v} : u \in \mathcal{N}_{r_1}(v)\}\!\!\})) : v \in [n]\}\!\!\}), \tag{35}$$

where $f(.; \theta)$ we assume that $f(.; \theta)$ is maximally expressive. The goal is to show that $f \sqsupseteq C(G; H)$. Using the transitivity of $\sqsupseteq$, we only need to choose appropriate $\phi, \psi, \varphi$ to achieve $\hat{f} = C(G; H)$ as the final representation. Let

$$\phi(\{\!\!\{z_v : v \in [n]\}\!\!\}) := \frac{1}{2 + 2 \times \mathbb{1}\{X_1^H = X_2^H\}} \sum_{i=1}^{n} z_i \tag{36}$$

$$\psi(X, (z, z')) := z \times \mathbb{1}\{X = X_1^H\} + z' \times \mathbb{1}\{X = X_2^H\} \tag{37}$$

$$\varphi(\{\!\!\{z_u : u \in [n']\}\!\!\}) := \Big( \sum_{i=1}^{n'} \mathbb{1}\{z_u = (X_2^H, 1)\}, \sum_{i=1}^{n'} \mathbb{1}\{z_u = (X_1^H, 1)\} \Big). \tag{38}$$

Then, a simple computation shows that

$$\hat{f}(G; \theta) = \phi(\{\!\!\{\psi(X_v^G, \varphi(\{\!\!\{X_u^{*v} : u \in \mathcal{N}_{r_1}(v)\}\!\!\})) : v \in [n]\}\!\!\}), \tag{39}$$
$$= C(G; H). \tag{40}$$

Since $\hat{f}(.; \theta)$ is an RNP-GNN with recursion parameter $r_1$ and for any maximally expressive RNP-GNN $f(.; \theta)$ with the same recursion parameter as $\hat{f}$ we have $f \sqsupseteq \hat{f}$ and $\hat{f} \sqsupseteq C(G; H)$, we conclude that $f \sqsupseteq C(G; H)$ and this completes the proof.

**Induction step.** Assume that the desired result holds for $t - 1$ ($t \geq 2$). We show that it also holds for $t$. Let us first define

$$\mathcal{H}^* := \{H_{v_1}^* : \exists v_2, \ldots, v_t \in [k] : (v_1, v_2, \ldots, v_t) \in \mathcal{C}_H(\mathbf{r})\} \tag{41}$$
$$c^*(H^*) := \mathbb{1}\{H^* \in \mathcal{H}^*\} \times \#\{v \in [k] : H_v^* \cong H^*\}. \tag{42}$$

Note that $\mathcal{H}^* \neq \emptyset$ by the assumption. Let

$$\|\mathcal{H}^*\| := \sum_{H^* \in \mathcal{H}^*} c^*(H^*). \tag{43}$$

For all $H^* \in \mathcal{H}^*$, using the induction hypothesis, there is a (universal) RNP-GNN $\hat{f}(.; \hat{\theta})$ with recursion parameters $(r_2, r_3, \ldots, r_t)$ such that $\hat{f} \sqsupseteq C(G; H^*)$. Using Proposition 4 we conclude

$$\hat{f} \sqsupseteq \sum_{u \in [k]: H_u^* \in \mathcal{H}^*} C(G; H_u^*). \tag{44}$$

Define a maximally expressive RNP-GNN with the recursion parameters $(r_1, r_2, \ldots, r_t)$ as follows:

$$f(G; \theta) = \phi(\{\!\!\{\psi(X_v^G, \hat{f}(G^*(\mathcal{N}_{r_1}(v)); \hat{\theta})) : v \in [n]\}\!\!\}). \tag{45}$$

Similar to the proof for $t = 1$, here we only need to propose a (not necessarily maximally expressive) RNP-GNN which achieves the function $C(G; H)$.

Let us define

$$f_{H_u^*}(G; \theta) := \phi(\{\!\!\{\psi_{H_u^*}(X_v^G, \xi \circ \hat{f}(G^*(\mathcal{N}_{r_1}(v)); \hat{\theta})) : v \in [n]\}\!\!\}), \tag{46}$$

where

$$\phi(\{\!\{z_v : v \in [n]\}\!\}) := \frac{1}{\|\mathcal{H}^*\|} \sum_{i=1}^{n} z_i \tag{47}$$

$$\psi_{H_u^*}(X, z) := z \times \mathbb{1}\{X = X_u^H\}, \tag{48}$$

$$\tag{49}$$

and $\xi \circ \hat{f} = C(G; H_u^*)$. Note that the existence of such function $\xi$ is guaranteed due to Proposition 2. Now we write

$$\|\mathcal{H}^*\| \times C(G; H) = \|\mathcal{H}^*\| \sum_{\mathcal{S} \subseteq [n]} \mathbb{1}\{G(\mathcal{S}) \cong H\} \tag{50}$$

$$= \sum_{\mathcal{S} \subseteq [n]} \sum_{v \in \mathcal{S}} \mathbb{1}\{\exists u \in [k] : (G(\mathcal{S} \setminus \{v\}))_v^* \cong H_u^* \in \mathcal{H}^* \wedge X_v^G = X_u^H\} \tag{51}$$

$$= \sum_{v \in [n]} \sum_{v \in \mathcal{S} \subseteq [n]} \mathbb{1}\{\exists u \in [k] : (G(\mathcal{S} \setminus \{v\}))_v^* \cong H_u^* \in \mathcal{H}^* \wedge X_v^G = X_u^H\} \tag{52}$$

$$= \sum_{v \in [n]} \sum_{v \in \mathcal{S} \subseteq \mathcal{N}_{r_1}(v)} \mathbb{1}\{\exists u \in [k] : (G(\mathcal{S} \setminus \{v\}))_v^* \cong H_u^* \in \mathcal{H}^* \wedge X_v^G = X_u^H\}$$
$$\tag{53}$$

$$= \sum_{v \in [n]} \sum_{v \in \mathcal{S} \subseteq \mathcal{N}_{r_1}(v)} \sum_{u \in [k] : H_u^* \in \mathcal{H}^*} \mathbb{1}\{(G(\mathcal{S} \setminus \{v\}))_v^* \cong H_u^*\}\mathbb{1}\{X_v^G = X_u^H\} \tag{54}$$

$$= \sum_{v \in [n]} \sum_{u \in [k] : H_u^* \in \mathcal{H}^*} C(G^*(\mathcal{N}_{r_1}(v)); H_u^*) \times \mathbb{1}\{X_v^G = X_u^H\}, \tag{55}$$

which means that

$$\sum_{u \in [k] : H_u^* \in \mathcal{H}^*} f_{H_u^*}(G; \theta) \sqsupseteq C(G; H). \tag{56}$$

However, for a maximally expressive RNP-GNN $f(.; \theta)$ we know that $f \sqsupseteq f_{H_u^*}$ for all $H_u^* \in \mathcal{H}$ and this means that $f \sqsupseteq C(G; H)$. The proof is thus complete.

## A.2 PROOF OF THEOREM 2

For any labeled graph $H$ on $r$ vertices (not necessarily connected) we claim that RNP-GNNs can count them.

**Claim 2.** *Let $f(.; \theta) : \mathbb{G}_n \to \mathbb{R}^d$ be a maximally expressive RNP-GNN with recursion parameters $(r - 1, r - 2, \ldots, 1)$. Then, $f \sqsupseteq C(G; H)$.*

Now consider the function

$$\ell(G) = \phi(\{\!\{\psi(G(\mathcal{S})) : \mathcal{S} \subseteq \mathcal{V}, |\mathcal{S}| \le r\}\!\}). \tag{57}$$

We claim that $f \sqsupseteq \ell$ ($f$ is defined in the previous claim) and this completes the proof according to Proposition 2.

To prove the claim, assume that $f(G_1) = f(G_2)$. Then, we conclude that $C(G_1; H) = C(G_2; H)$ for any labeled $H$ (not necessarily connected) with $r$ vertices. Now, we have

$$\ell(G) = \phi(\{\!\{\psi(G(\mathcal{S})) : \mathcal{S} \subseteq \mathcal{V}, |\mathcal{S}| \le r\}\!\}) \tag{58}$$
$$= \phi(\{\!\{\psi(H) : H \in \mathbb{G}_r, \text{ the multiplicity of } H \text{ is } C(G; H)\}\!\}), \tag{59}$$

which shows that $\ell(G_1) = \ell(G_2)$.

*Proof of Claim 2.* To prove the claim, we use an induction on the number of connected components $c_H$ of graph $H$. If $H$ is connected, i.e., $c_H = 1$, then according to Theorem 1, we know that $f \sqsupseteq C(G; H)$.

Now assume that the claim holds for $c_H = c - 1 \geq 1$. We show that it also holds for $c_H = c$. Let $H_1, H_2, \ldots, H_c$ denote the connected components of $H$. Also assume that $H_i \not\cong H_j$ for all $i \neq j$. We will relax this assumption later. Let us define

$$\mathcal{A}_G := \{(\mathcal{S}_1, \mathcal{S}_2, \ldots, \mathcal{S}_c) : \forall i \in [c] : \mathcal{S}_i \subseteq [n]; G(\mathcal{S}_i) \cong H_i\}. \tag{60}$$

Note that we can write

$$|\mathcal{A}_G| = \prod_{i=1}^{c} C(G; H_i) \tag{61}$$

$$= C(G; H) + \sum_{j=1}^{\infty} c'_j C(G; H'_j), \tag{62}$$

where $H'_1, H'_2, \ldots$ are all non-isomorphic graphs obtained by adding edges (at least one edge) between $c$ graphs $H_1, H_2, \ldots, H_c$, or contracting a number of vertices of them. The constants $c'_j$ are just used to remove the effect of multiple counting due to the symmetry. Now, since for any $H_i, H'_j$ the number of connected components is strictly less that $c$, using the induction, we have $f \sqsupseteq C(G; H_i)$ and $f \sqsupseteq C(G; H'_j)$ for all $j$ and all $i \in [c]$. According to Proposition 4, we conclude that $f \sqsupseteq C(G; H)$ and this completes the proof. Also, if $H_i$, $i \in [c]$, are not pairwise non-isomorphic, then we can use $\alpha C(G; H)$ in above equation instead of $C(G; H)$, where $\alpha > 0$ removes the effect of multiple counting by symmetry. The proof is thus complete.

### A.3 PROOF OF THEOREM 3

To prove Theorem 3, we need to bound the number of node updates required for an RNP-GNN with recursion parameters $(r_1, r_2, \ldots, r_t)$. First of all, we have $n$ variables used for the final representations of vertices. For each vertex $v_1 \in \mathcal{V}$, we explore the local neighborhood $\mathcal{N}_{r_1}(v_1)$ and apply a new RNP-GNN network to that neighborhood. In other words, for the second step we need to update $|\mathcal{N}_{r_1}(v_1)|$ nodes. Similarly, for the $i$th step of the algorithm we have as most

$$\lambda_i := \max_{v_1 \in [n]} \max_{\substack{v_{j+1} \in \mathcal{N}_{r_j}(v_j) \\ \forall j \in [i-1]}} |\mathcal{N}_{r_1}(v_1) \cap \mathcal{N}_{r_2}(v_2) \cap \mathcal{N}_{r_3}(v_3) \ldots \cap \mathcal{N}_{r_i}(v_i)|, \tag{63}$$

updates. Therefore, we can bound the number of node updates as

$$n \times \prod_{i=1}^{t} \lambda_i. \tag{64}$$

Since $\lambda_i$ is decreasing in $i$, we simply conclude the desired result.

### A.4 PROOF OF THEOREM 4

Let $K_k$ denote the complete graph on $k$ vertices.

**Claim 3.** *For any $k, n \in \mathbb{N}$, such that $n$ is sufficiently large,*

$$\left|\{C(G; K_k) : G \in \mathbb{G}_n\}\right| \geq \frac{(cn/k \log(n/k) - k)^k}{k!} = \tilde{\Omega}(n^k), \tag{65}$$

*where $c$ is a constant which does not depend on $k, n$.*

In particular, we claim that the number of different values that $C(G; K_k)$ can take is $n^k$, up to poly-logarithmic factors.

To prove the theorem, we use the above claim. Consider a class of $(s, t)-$good graph representations $f(.; \theta)$ which can count any substructure on $k$ vertices. As a result, $f \sqsupseteq C(G; K_k)$ for an appropriate parametrization $\theta$. By the definition, $f(.)$ must take at least $\left|\{C(G; K_k) : G \in \mathbb{G}_n\}\right|$ different values, i.e.,

$$\left|\{f(G; \theta) : G \in \mathbb{G}_n\}\right| \geq \left|\{C(G; K_k) : G \in \mathbb{G}_n\}\right|. \tag{66}$$

Also,

$$\left|\{f(G;\theta) : G \in \mathbb{G}_n\}\right| \leq \left|\{\!\{\psi(G_i) : i \in [t]\}\!\} : G \in \mathbb{G}_n\}\right|, \tag{67}$$

where $(G_1, G_2, \ldots, G_t) = \Xi(G)$. But, $\psi$ can take only $s$ values. Therefore, we have

$$\left|\{C(G;K_k) : G \in \mathbb{G}_n\}\right| \leq \left|\{f(G;\theta) : G \in \mathbb{G}_n\}\right| \tag{68}$$

$$\leq \left|\{\!\{\psi(G_i) : i \in [t]\}\!\} : G \in \mathbb{G}_n\}\right| \tag{69}$$

$$\leq \left|\{\!\{\alpha_i : i \in [t]\}\!\} : \forall i \in [t] : \alpha_i \in [s]\}\right| \tag{70}$$

$$\leq (t+1)^{s-1}. \tag{71}$$

As a result, $(t+1)^{s-1} = \tilde{\Omega}(n^k)$ or $t = \tilde{\Omega}(n^{\frac{k}{s-1}})$. To complete the proof, we only need to prove the claim.

*Proof of Claim 3.* Let $p_1, p_2, \ldots, p_m$ be distinct prime numbers less than $n/k$. Using the prime number theorem, we know that $\lim_{n \to \infty} \frac{m}{n/k \log(n/k)} = 1$. In particular, we can choose $n$ large enough to ensure $cn/k \log(n/k) < m$ for any constant $c < 1$.

For any $\mathcal{B} = \{b_1, b_2, \ldots, b_k\} \subseteq [m]$, define $G_\mathcal{B}$ as a graph on $n$ vertices such that $\mathcal{V}_{G_\mathcal{B}} = V_0 \sqcup (\sqcup_{i \in [k]} \mathcal{V}_i)$, and $|\mathcal{V}_i| = p_{b_i}$. Also,

$$e = (u, v) \in G_\mathcal{B} \iff \exists i, j \in [m], i \neq j : u \in \mathcal{V}_i \ \& \ v \in \mathcal{V}_j. \tag{72}$$

The graph $G_\mathcal{B}$ is well-defined since $\sum_{i=1}^k p_{b_i} \leq k \times n/k = n$. Note that $C(G_\mathcal{B}; K_k) = \prod_{i=1}^k p_{b_i}$. Also, since $p_i, i \in [m]$, are prime numbers, there is a unique bijection

$$\mathcal{B} \xleftrightarrow{\varphi} C(G_\mathcal{B}; K_k). \tag{73}$$

Therefore,

$$\left|\{C(G;K_k) : G \in \mathbb{G}_n\}\right| \geq \left|\{C(G_\mathcal{B}; K_k) : \mathcal{B} \subseteq [m], |\mathcal{B}| = k\}\right| \tag{74}$$

$$= \binom{m}{k} \tag{75}$$

$$\geq \frac{(m-k)^k}{k!} \tag{76}$$

$$\geq \frac{(cn/k \log(n/k) - k)^k}{k!}. \tag{77}$$

## B   RELATIONSHIP TO THE RECONSTRUCTION CONJECTURE

Theorem 2 provides a universality result for RNP-GNNs. Here, we note that the proposed method is closely related to the reconstruction conjecture, an old open problem in graph theory. This motivates us to explain their relationship/differences. First, we need a definition for unlabeled graphs.

**Definition 10.** *Let $\mathcal{F}_n \subseteq \mathbb{G}_n$ be a set of graphs and let $G_v = G(\mathcal{V} \setminus \{v\})$ for any finite simple graph $G = (\mathcal{V}, \mathcal{E})$, and any $v \in \mathcal{V}$. Then, we say the set $\mathcal{F}$ is reconstructible if and only if there is a bijection*

$$\{\!\{G_v : v \in \mathcal{V}\}\!\} \xleftrightarrow{\Phi} G, \tag{78}$$

*for any $G \in \mathcal{F}_n$. In other words, $\mathcal{F}_n$ is reconstructible, if and only if the multi-set $\{\!\{G_v : v \in \mathcal{V}\}\!\}$ fully identifies $G$ for any $G \in \mathcal{F}_n$.*

It is known that the class of disconnected graphs, trees, regular graphs, are reconstructible (Kelly et al., 1957; McKay, 1997). The general case is still open; however it is widely believed that it is true.

**Conjecture 1** (Kelly et al. (1957))**.** $\mathbb{G}_n$ *is reconstructible.*

For RNP-GNNs, the reconstruction from the subgraphs $G_v^*$, $v \in [n]$ is possible, since we relabel any subgraph (in the definition of $X^*$) and this preserves the critical information for the recursion to the original graph. In the reconstruction conjecture, this part of information is missing, and this makes the problem difficult. Nonetheless, since in RNP-GNNs we preserve the original node's information in the subgraphs with relabeling, the reconstruction conjecture is not required to hold to show the universality results for RNP-GNNs, although that conjecture is a motivation for this paper. Moreover, if it can be shown that the reconstruction conjecture it true, it may be also possible to find a simple encoding of subgraphs to an original graph and this may lead to more powerful but less complex new GNNs.

## C   THE RNP-GNN ALGORITHM

In this section, we provide pseudocode for RNP-GNNs. The algorithm below computes node representations. For a graph representation, we can aggregate them with a common readout, e.g., $h_G \leftarrow \mathrm{MLP}\left( \sum_{v \in \mathcal{V}} h_v^{(t)} \right)$. Following (Xu et al., 2019), we use sum pooling here, to ensure that we can represent injective aggregation functions.

---

**Algorithm 1** Recursive Neighborhood Pooling-GNN (RNP-GNN)

---

**Input:** $G = (\mathcal{V}, \mathcal{E}, \{x_v\}_{v \in \mathcal{V}})$ where $\mathcal{V} = [n]$, recursion parameters $r_1, r_2, \ldots, r_t \in \mathbb{N}$, $\epsilon^{(i)} \in \mathbb{R}$, $i \in [t]$, node features $\{x_v\}_{v \in \mathcal{V}}$.
**Output:** $h_v$ for all $v \in \mathcal{V}$
 $h_v^{\mathrm{in}} \leftarrow x_v$ for all $v \in \mathcal{V}$
 **if** $t = 1$ **then**

$$h_v \leftarrow \mathrm{MLP}^{(t)}\left( (1 + \epsilon^{(1)}) h_v^{\mathrm{in}} + \sum_{u \in \mathcal{N}_{r_1}(v) \setminus \{v\}} (h_u^{\mathrm{in}}, \mathbb{1}(u, v) \in \mathcal{E})' \right),$$

  for all $v \in \mathcal{V}$.
 **else**
  **for** all $v \in V$ **do**
   $G_v' \leftarrow G(\mathcal{N}_{r_1}(v) \setminus \{v\})$ with node features $\{h_u^{\mathrm{in}}\}$
   $\{\hat{h}_{v,u}\}_{u \in G_v' \setminus \{v\}} \leftarrow \mathrm{RNP\text{-}GNN}(G_v', (r_2, r_3, \ldots, r_t), (\epsilon^{(2)}, \ldots, \epsilon^{(t)}))$
   $h_v \leftarrow \mathrm{MLP}^{(t)}\left( (1 + \epsilon^{(t)}) h_v^{\mathrm{in}} + \sum_{u \in \mathcal{N}_{r_1}(v) \setminus \{v\}} \hat{h}_{u,v} \right)$.
  **end for**
 **end if**
 **return** $\{h_v\}_{v \in \mathcal{V}}$

---

With this algorithm, one can achieve the expressive power of RNP-GNNs if high dimensional MLPs are allowed (Xu et al., 2019; Hornik et al., 1989; Hornik, 1991). That said, in practice, smaller MLPs may be acceptable (Xu et al., 2019).

## D   COMPUTING A COVERING SEQUENCE

As we explained in the context of Theorem 1, we need a covering sequence (or an upper bound to that) to design an RNP-GNN network that can count a given substructure. A covering sequence can be constructed from a spanning tree of the graph.

For reducing complexity, it is desirable to have a covering sequence with minimum $r_1$ (Theorem 3). Here, we suggest an algorithm for obtaining such a covering sequence, shown in Algorithm 2. For obtaining merely an aribtrary covering sequence, one can compute any minimum spanning tree (MST), and then proceed as with the MST in Algorithm 2.

Given an MST, we build a vertex covering sequence by iteratively removing a leave $v_i$ from the tree and adding the respective node $v_i$ to the sequence. This ensures that, at any point, the remaining graph is connected. At position $i$ corresponding to $v_i$, the covering sequence contains the maximum

distance $r_i$ of $v_i$ to any node in the remaining graph, or an upper bound on that. For efficiency, an upper bound on the distance can be computed in the tree.

To minimize $r_1 = \max_{u \in \mathcal{V}} d(u, v_1)$, we need to ensure that a node in $\arg\min_{v \in \mathcal{V}} \max_{u \in \mathcal{V}} d(u, v)$ is a leaf in the spanning tree. Hence, we first compute $\max_{u \in \mathcal{V}} d(u, v)$ for all nodes $v$, e.g., by running All-Pairs-Shortest-Paths (APSP) (Kleinberg & Tardos, 2006), and sort them in increasing order by this distance. Going down this list, we try whether it is possible to use the respective node as $v_1$, and stop when we find one.

Say $v^*$ is the current node in the list. To compute a spanning tree where $v^*$ is a leaf, we assign a large weight to all the edges adjacent to $v^*$, and a very low weight to all other edges. If there exists such a tree, running an MST with the assigned weights will find one. Then, we use $v^*$ as $v_1$ in the vertex covering sequence. This algorithm runs in polynomial time.

---

**Algorithm 2** Computing a covering sequence with minimum $r_1$

---

**Input:** $H = (\mathcal{V}, \mathcal{E}, X)$ where $\mathcal{V} = [t+1]$
**Output:** A minimal covering sequence $(r_1, r_2 \ldots, r_t)$, and its corresponding vertex covering sequence $(v_1, v_2, \ldots, v_{t+1})$
    For any $u, v \in \mathcal{V}$, compute $d(u, v)$ using APSP
    $(u_1, u_2, \ldots, u_{t+1}) \leftarrow$ all the vertices sorted increasingly in $s(v) := \max_{u \in \mathcal{V}} d(u, v)$
    **for** $i = 1$ to $t + 1$ **do**
        Set edge weights $w(u, v) = 1 + t \times \mathbb{1}\{u = u_i \vee v = u_i\}$ for all $(u, v) \in \mathcal{E}$
        $H_T \leftarrow$ the MST of $H$ with weights $w$
        **if** $u_i$ is a leaf in $H_T$ **then**
            $v_1 \leftarrow u_i$
            $r_1 \leftarrow s(u_i)$
            **break**
        **end if**
    **end for**
    **for** $i = 2$ to $t + 1$ **do**
        $v_i \leftarrow$ one of the leaves of $H_T$
        $r_i \leftarrow \max_{u \in \mathcal{V}_{H_T}} d(u, v_i)$
        $H_T \leftarrow H_T$ after removing $v_i$
    **end for**
    **return** $(r_1, r_2, \ldots, r_t)$ and $(v_1, v_2, \ldots, v_{t+1})$

---

