# OpenReview forum: "Recursive Neighborhood Pooling for Graph Representation Learning"
_ICLR.cc/2021/Conference — Reject_

### Official Review · AnonReviewer2 · 2020-10-28
**Interesting work from theoretical perspective but without real world applications.**

**Rating:** 6
**Confidence:** 1

**Review:**

The proposed paper seeks a theoretical possibility of counting the subgraph by a graph neural network. To this end, the authors proposed a recursive neighborhood pooling graph neural network and proved the express power of the model. The universal approximation results on a subgraph have been shown as well. Analysis of computational complexity shows the algorithm is much efficient than the known class of models that can count substructures.

The strength of this paper is the detailed theoretical analysis of the proposed method. The analysis has been taken from multiple perspectives: model complexity, computational complexity, and theoretic lower-bound on the class of the proposed method. The paper gives a new insight into measuring the representation power of graph neural networks through subgraph counting.

Although subgraph counting is an interesting problem from a theoretical viewpoint, the weakness of this paper is that it is not answered how relevant it is to real-world tasks and how successful it is. Since this is the first paper proposing a new way of measuring the express power of the graph neural network, it would have been much better if there's some discussion on this point.

---

> ### Author Response · Authors · 2020-11-25
> **Thank you for reading the paper and for your feedback.**
>
>
> Thank you for reading the paper and for your feedback. Below we address your questions.
>
> Q: Relevance of subgraph counting
>
> A: In general, given the great interest in graph representation learning, the question of characterizing the tradeoffs between the expressive power and the computational complexity of graph neural networks is of interest to the community, as shown by several recent works on higher-order GNNs, e.g., references (Morris et al., 2019; Maron et al., 2019; Chen et al. (2020)) in the paper, and the open question [1].
> In this paper, we address this question for the problem of counting subgraphs. The counting problem is motivated by recent works showing that low-order message passing GNNs cannot learn it [2,3]. Counting subgraphs is important, for instance, in computational chemistry, drug design and materials science, to identify functional groups in molecules that are relevant for properties and reactivity. In addition, recent works in that area (e.g. [4]) have used motifs (subgraphs) as building blocks.
>
> [1] H. Maron, H. Ben-Hamu, Y. Lipman. Approximation Power of Invariant Graph Networks. NeurIPS 2019 Workshop on Graph Representation Learning.
>
> [2] Z. Chen, L. Chen, S. Villar, J. Bruna. Can graph neural networks count substructures?  NeurIPS 2020.
>
> [3] V. Garg, S. Jegelka, T. Jaakkola. Generalization and representational limits of graph neural networks. ICML 2020.
>
> [4] W. Jin, R. Barzilay, T. Jaakkola. Hierarchical Generation of Molecular Graphs using Structural Motifs. ICML 2020.

---

### Official Review · AnonReviewer4 · 2020-10-28
**Theoretical results on representative power of graph neural networks leveraging an efficient recursive procedure to assemble higher-order neighbourhoods of a node**

**Rating:** 6
**Confidence:** 4

**Review:**

The paper has a fully theoretical flair while proposing a novel and seemingly efficient procedure to recursively compute the higher-order (i.e. more than 1-hop) neighbourhood of a node that are used for learning discriminative graph embedding. The paper contributes with the model above (RNP-GNN) and by providing a proof of its representational power and a general theorem supplying an information theoretic lower bound on the complexity of GNNs that can count induced substructures.

To the extent of my knowledge the recursive neighbourhood construction procedure presented in the paper is original and it is based on solid graph-theoretic concepts. A proof of the asymptotic complexity of the approach is provided in Section 6, suggesting that the approach can yield to increased representational power at minor computational costs. These results are derived under a sparsity assumption which is not completely clear how much it would hold in practice. In particular, I am not sure whether the sparsity constraint should hold on average in the graph or if the presence of a single node infringing the constraint will induce an exponential growth in complexity. I am thinking in particular to social graphs where hubness might become an issue if the second is true. I would like to read some elaborations as concerns this specific point.

The theoretical results appear solid and the proof well-constructed although, admittingly, I have not checked the proofs in Appendix in detail. I do have one question as regards the construction of counting substructures proof. Since it entails leveraging vertex covers, does it require this to be minimal or any vertex cover would suffice? Is there any change if the cover is not minimal?
Concluding, while the paper has certainly good quality theoretical contributions, it lacks an empirical analysis (even a small one, given the theoretical nature of the problem), which would have made the paper claims more convincing. In particular, since much of the results hang on the tradeoff between representational power and efficiency, one would have liked to see an empirical proof of such excellent tradeoff, possibly in graphs of different nature (i.e. bio-chemical and social, given that they tend to have considerably different connectivity patterns). The short but informative empirical analysis in the paper by Murphy et al, ICML 2019 is an excellent example.

One final remark: the paper, while generally well written, has misprints here and there which can be easily spotted by careful proofreading. Not listing them here, but they should be taken care of.

=========== POST REBUTTAL

The rebuttal is very helpful and "to the point" in clarifying the issues that I had raised as concerns the impact of the assumptions taken in the theoretical proofs. The fact that complexity hinges on average node degree and that any vertex cover sufficies for the proof confirms that the approach put forward in the paper might work out of the theoretical box. The Authors also suggest that the theoretical framework can be translated to a running model with a certain ease and that, in fact, its practical implementation and empircal assessment is on the way. Which brings me to the key point in my assessment. I am convinced there is value in this work and in the theoretical contribution in the paper.  I am not convinced that this paper can have a strong impact without an empirical validation. As I have underlined in my review, there are several related works in literature, with a similar theoretical flair which, nevertheless, provided at least a simple empirical validation. I believe that this paper shold do the same: it would be stronger, more complete and with a higher potential to influence the community. As it is, this is a borderline paper (leaning on the positive side).

---

> ### Author Response · Authors · 2020-11-25
> **Thank you very much for your detailed reading, feedback and support.**
>
> Thank you very much for your detailed reading, feedback and support. Here, we provide replies to the questions.
>
> Q: What exactly is the sparsity assumption, and how would one node with large degree affect the complexity?
>
> A: Indeed, one of the theoretical benefits of RNP-GNNs is that for sparse graphs, they only need a close to linear number of node updates. The definition of ``sparse'' here is a restriction on the maximum degree of all nodes in the graph. If the graph has only few nodes with larger degree, we may have two cases:
>
> 1) The local neighborhood sizes collapse after a few iterations, then the overall complexity is still better than the worst case complexity.
>
> 2) The local neighborhood sizes remain large. In fact, if the limitation is on the average degree (not the max degree), then, in general, it is impossible to guarantee anything better. To clarify this observation and put it in context, we added a discussion of existing complexity results, and relate them to our results (Section 8). In particular, recent results in complexity (Gishboliner et al. (2020); Bera et al. (2019)) show that if only the average degree of the input graph is constant, e.g., if we only have a few vertices with large degree, then no algorithm can count substructures in guaranteed linear time. This implies that our results are close to optimal.
>
>
>
> Q: Would any vertex cover suffice for the proof?
>
> A: For Theorem 1, any valid vertex covering sequence works. Having a minimum $r_1$ is not required.
> We clarified this in the revised paper, and also added a simple algorithm for obtaining a covering sequence from a spanning tree of the graph, in Appendix D: 1) Compute a spanning tree, e.g., an MST in time $O(n \log(n))$. 2) Construct a vertex covering sequence by iteratively picking a leaf of the tree, adding it to the sequence, and removing it from the tree. This way, the remaining graph is guaranteed to remain connected. 3) To obtain a valid covering sequence for this vertex covering sequence, each time one selects a leaf, one computes the farthest distance to any neighbor in the tree (this is an upper bound but valid). If we want the covering sequence with minimum $r_1$, then we first search for the node that has the smallest distance to its farthest neighbor, and enforce that to be the first leaf to be pruned.
>
>
>
> Q: Empirical results
>
> A: We agree that empirical results would be valuable, and are currently working on experiments that we may add to the final version of the paper.
>  That said, we believe that even as is, the insights provided by the paper are interesting for the community. Given the great interest in graph representation learning, the question of characterizing the tradeoffs between the expressive power and the computational complexity of graph neural networks has driven several recent works on higher-order GNNs, and an open question. In this paper, we address this question for the problem of counting subgraphs, and show results that are close to optimal.
>
> Q: Typos
>
> A: Thank you for pointing these out. We corrected them in the revised version.

---

### Official Review · AnonReviewer1 · 2020-10-28
**Review of Recursive Neighborhood Pooling for Graph Representation Learning**

**Rating:** 6
**Confidence:** 4

**Review:**

Summary:
The goal of the paper is to show that GNN's (without exponential computational complexity) can be constructed with the ability to count subgraphs. To this effect, the authors propose a principled neighborhood pooling strategy and theoretically characterize their expressive power - with respect to other models proposed earlier. More specifically, the authors propose a recursive neighborhood pooling strategy which characterizes graphs based on the counts of subgraphs . Furthermore, they show that if the tuple of recursion parameters are chosen well, their proposed model can capture all induced subgraphs (universality) of sizes smaller than the first value in the tuple of recursion parameters plus 1 - and show a relationship to the reconstruction conjecture (Kelly et al. 1957). The authors also provide a bound on the number of iterations required to learn the expressive representations.

Pros:
1. Theoretical contributions which provide a strategy to learn expressive representations with the ability to count subgraphs and distinguish graphs based on the same.
2. Proposed model provides a strategy to perform a tradeoff on the exponential nature required to compute sugraphs in earlier works.
3. Comprehensive literature review

Concerns:
1. The theorems all state that, there exists a set of recursion parameters, etc - but do not provide a computationally tractable strategy to determine the same for real world graphs (e.g. diameters of the graph > 15) - or a strategy to determine the vertex covering sequence if it exists - which is the basis for theorem 1(among all permutations of the set $\mathcal{S}$ - without looking over all permutations - this can explode even with subgraphs of size 10 for instance).
2. Lack of any experimental evidence to empirically show gains - While the authors explicitly say that their work is theoretical, they could have included synthetic experiments to demonstrate the same- given that the Local Relational Pooling,  k-WL works and k-IGN (the latter two, which are exponential) works have experimental results.

If the concerns are addressed, I will be happy to update my scores.

---

> ### Author Response · Authors · 2020-11-25
> **Thank you very much for your detailed reading, feedback and support.**
>
> Thank you very much for your detailed reading, feedback and support. Here, we provide replies to the questions.
>
> Q: How can we compute covering sequences?
>
> A: Thank you for pointing this out, we clarified this question by adding an algorithm for computing a covering sequence in Appendix D, and now summarize the strategy in the main paper too. One simple way to obtain a covering sequence for a given graph is as follows: 1) Compute a spanning tree, e.g., an MST in time $O(n \log(n))$. 2) Construct a vertex covering sequence by iteratively picking a leaf of the tree, adding it to the sequence, and removing it from the tree. This way, the remaining graph is guaranteed to remain connected. 3) To obtain a valid covering sequence for this vertex covering sequence, each time one selects a leaf, one computes the farthest distance to any neighbor in the tree (this is an upper bound but valid). The theorem works with any valid covering sequence. If we want the covering sequence with minimum $r_1$, then we first search for the node that has the smallest distance to its farthest neighbor, and enforce that to be the first leaf to be pruned.
>
> Q: Empirical results
>
> A: We agree that an empirical results would be valuable, and are currently working on experiments that we may add to the final version of the paper.
>  That said, we believe that even as is, the insights provided by the paper are interesting for the community. Given the great interest in graph representation learning, the question of characterizing the tradeoffs between the expressive power and the computational complexity of graph neural networks has driven several recent works on higher-order GNNs, and an open question. In this paper, we address this question for the problem of counting subgraphs, and show results that are close to theoretical lower bounds.

---

### Official Review · AnonReviewer3 · 2020-10-29
**Interesting idea, with nice theoretical results, but seems rather hypothetical with no demonstration that the theory can feasibly be implemented and no indication it really has impact in practice**

**Rating:** 4
**Confidence:** 4

**Review:**

**Update following author response and reviewer discussion:**

I would like to thank the authors for providing a response, and in particular for providing further justification for their injectivity assumption. However, the main concern remains the lack of empirical validation, even on toy examples, showing that (or whether) the derived theory here goes beyond a hypothetical thought exercise and can be implemented in practice. It shouldn't be difficult to provide such examples, assuming the proposed approach does indeed work as indicated by the theory developed here, and without such examples, the paper seems rather incomplete. Therefore, unfortunately, my score remains unchanged at this point, although I would like to encourage the authors to keep pursuing this direction.

---

**Initial review:**

This manuscript discusses a hypothetical pooling approach that recursively applies a generic (unspecified) GNN to subgraph constructed from local neighborhoods of a certain radius (or sequence of them, applied in successive applications of the recursion) in an input graph. The main motivation set out by this work is to prove that, under some assumptions (which may or may not be realistic) on the applied GNNs, this pooling strategy is sufficiently expressive to admit a configuration of weights that would enable counting subgraphs of certain size in the main input graph. The authors provide theoretical study of this expressivity and prove the capability to count subgraphs in an existence sense. That is, they show there exists some function that can be written in the form of their proposed pooling architecture (with sufficiently rich architecture between pooled levels) such that non isomorphic subgraphs would necessarily be distinguishable by some of the extracted features, and thus counting equivalency classes of these features would provide the desired subgraph counting. The authors also provide other results showing functions written in the form of their pooling based architecture have reasonable complexity of node operations, and providing some lower bound on the capability of any GNN approach to count subgraphs, thus putting their theoretical results in context.

The proposed pooling approach itself is rather straight forward, but intuitively makes sense. Given an input sequence of radii, at each iteration of the recursion each node in the graph constructs a subgraph over nodes that are within the current considered radius in the sequence. Then, by recursion, the same approach is applied to each of these subgraphs, with the remainder of the radii sequence (subsequent to the current one) being passed on, until reaching only a single radius or a single node as stopping conditions. Given the resulting (sub)graph features for each node, these are combined with the input node features via generic functions parametrized by unspecified neural networks (written as an MLP with arbitrary architecture for the sake of presenting pseudocode in the supplement) that are supposed to be injective, when carefully examining the derivation and proofs in the supplement. Finally, the resulting node features are aggregated together (again, via generic functions modeled as unspecified neural networks that are supposed to be injective, this time up to node permutation) into graph features that provide a readout to be returned as the output of the network to outer levels of the recursion, or finally to the general output of the network.

It seems to me that the main theoretical results here crucially hinge on the injectivity assumption, and yet the authors make no attempt to verify that any existing GNN indeed provides an architecture that satisfies their assumed conditions. More importantly, they do not provide any empirical evidence that the approach described here is viable in practice. There is no implementation or demonstration of this approach, even on toy examples, let alone benchmarks on graph datasets. There is also no discussion on the impact of the proposed pooling on the training of the resulting RNP-GNN. It is not clear to me that this approach is feasible to apply in practice, beyond a hypothetical thought experiment. Therefore, while this line of research seems like an interesting direction, I find it somewhat lackluster and do not thing it is mature enough at this stage to be accepted for publication.

Finally, as a minor remark, the writing here seems somewhat rushed, with several typos and unclear phrases. Some examples:
- Equation (4) should probably have "RNP-GNN" rather than "RNN-GNN"
- The last line before section 5: "Appendix ??" missing a proper reference.
- Page 6: "sincethe" missing a space between "since" and "the"
- Page 7: "used for for aggregation" has a duplicated "for"
- The last sentence of section 5 (right before section 6) should be rephrased - I couldn't decipher it grammatically.
There may be more that I missed, but the paper and supplement would benefit from some additional proofreading.

---

> ### Author Response · Authors · 2020-11-25
> **Thank you very much for your detailed reading and feedback.**
>
> Thank you very much for your detailed reading and feedback. Here, we provide replies to the raised concerns:
>
> Q: Injectivity assumption
>
> A: The injectivity assumption builds on prior work. For example, [1] identifies injectivity as an important property, and proposes an architecture with injective aggregations, Graph Isomorphism Networks (GINs), which uses MLPs and sum aggregations. The experiments in that paper show that the sum makes a difference, even with relatively small MLPs. So, in our case, one could use the same aggregations. Other works that use injectivity are Invariant Graph Networks (IGNs) [2],  and k-Graph Neural Networks (k-GNNs) [3]. Hence, this is a common assumption that is sufficiently realizable in practice.
>
> [1]  K. Xu, W. Hu, J. Leskovec, S. Jegelka. How powerful are graph neural networks? ICLR, 2019.
>
> [2]  H. Maron, E. Fetaya, N. Segol, Y. Lipman. On the Universality of Invariant Networks, ICML, 2019.
>
> [3] C. Morris, M. Ritzert, M. Fey, W. L Hamilton, J. E. Lenssen, G. Rattan, M. Grohe. Weisfeiler and leman go neural: Higher-order graph neural networks. AAAI, 2019.
>
> Q: Empirical evaluation
>
> A: We agree that empirical results would be valuable, and are currently working on experiments that we may add to the final version of the paper.
> That said, we believe that even as is, the insights provided by the paper are interesting for the community. Given the great interest in graph representation learning, the question of characterizing the tradeoffs between the expressive power and the computational complexity of graph neural networks has driven several recent works on higher-order GNNs, e.g., references (Morris et al., 2019; Maron et al., 2019; Chen et al. (2020)) in the paper, and the open question [4].
> In this paper, we address this problem for the problem of counting subgraphs, motivated by recent works (Chen et al NeurIPS 2020, Garg et al ICML 2020) and its relevance for representing e.g.\ tasks in computational chemistry and drug design. We provide a new model that, on subclasses of graphs, achieves up to exponential improvements in complexity over existing models. We also provide lower bounds that show this result is close to optimal. To illustrate this point further, we added a discussion (Section 8) that relates our results, and the problem of subgraph counting, to the bigger context of known results in complexity, demonstrating the tightness of our results. These results characterize the sought tradeoffs.
>
> [4] H. Maron, H. Ben-Hamu, Y. Lipman. Approximation Power of Invariant Graph Networks. NeurIPS 2019 Workshop on Graph Representation Learning.
>
>
> Q: Typos
>
> A: Thank you very much for pointing out the typos. We fixed them in the revision.

---

### Decision · Program_Chairs · 2021-01-07
**Final Decision**

**Decision:**

Reject

**Comment:**

The reviewers agree that this is an interesting and promising paper, although it is on the theoretical side, without even satisfying toy examples to demonstrate its usefulness. This itself is not a fatal problem (ICLR can and should welcome theoretical papers), however including such experiments would significantly strengthen the impact of this paper, and make it more competitive with other ICLR submissions.